# A Medical Data-Effective Learning Benchmark for Highly Efficient Pre-training of Foundation Models

### Wenxuan Yang[*]
Shanghai Key Laboratory of Intelligent Information Processing, School of Computer Science, Fudan University
Shanghai, China
wxyang23@m.fudan.edu.cn

### Weimin Tan[*]
Shanghai Key Laboratory of Intelligent Information Processing, School of Computer Science, Fudan University
Shanghai, China
wmtan@fudan.edu.cn

### Yuqi Sun
Shanghai Key Laboratory of Intelligent Information Processing, School of Computer Science, Fudan University
Shanghai, China
yqsun22@m.fudan.edu.cn

### Bo Yan[†]
Shanghai Key Laboratory of Intelligent Information Processing, School of Computer Science, Fudan University
Shanghai, China
byan@fudan.edu.cn

## Abstract

Foundation models, pre-trained on massive datasets, have achieved unprecedented generalizability. However, is it truly necessary to involve such vast amounts of data in pre-training, consuming extensive computational resources? This paper introduces data-effective learning, aiming to use data in the most impactful way to pre-train foundation models. This involves strategies that focus on data quality rather than quantity, ensuring the data used for training has high informational value. Data-effective learning plays a profound role in accelerating foundation model training, reducing computational costs, and saving data storage, which is very important as the volume of medical data in recent years has grown beyond many people's expectations. However, due to the lack of standards and comprehensive benchmark, research on medical data-effective learning is poorly studied. To address this gap, our paper introduces a comprehensive benchmark specifically for evaluating data-effective learning in the medical field. This benchmark includes a dataset with millions of data samples from 31 medical centers (DataDEL), a baseline method for comparison (MedDEL), and a new evaluation metric (NormDEL) to objectively measure data-effective learning performance. Our extensive experimental results show the baseline MedDEL can achieve performance comparable to the original large dataset with only 5% of the data. Establishing such an open data-effective learning benchmark is crucial for the medical foundation model research community because it facilitates efficient data use, promotes collaborative breakthroughs, and fosters the development of cost-effective, scalable, and impactful healthcare solutions. The benchmark can be accessed at **GitHub Repository**.

[*]Wenxuan Yang and Weimin Tan contributed equally to this work and should be considered co-first authors. Bo Yan is the corresponding author.

## CCS Concepts

• **Computing methodologies → Computer vision**.

## Keywords

Medical benchmark, Data-Effective learning, Endoscopic Image Processing, Foundation Model

**ACM Reference Format:**
Wenxuan Yang, Weimin Tan, Yuqi Sun, and Bo Yan. 2024. A Medical Data-Effective Learning Benchmark for Highly Efficient Pre-training of Foundation Models. In *Proceedings of the 32nd ACM International Conference on Multimedia (MM '24), October 28-November 1, 2024, Melbourne, VIC, Australia.* ACM, New York, NY, USA, 10 pages. https://doi.org/10.1145/3664647.3681313

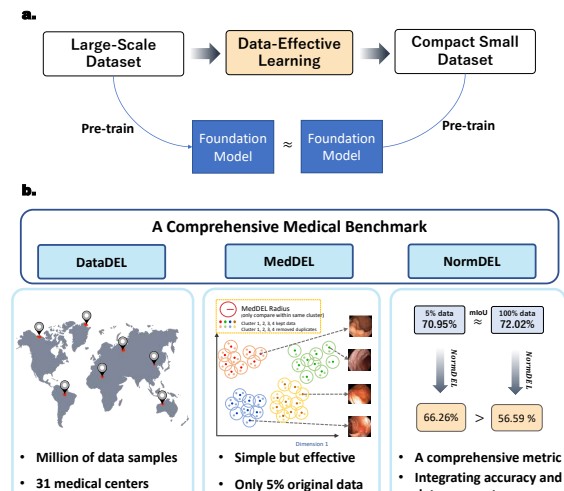

**Figure 1: Data-Effective Learning (DEL) enables more efficient pre-training of foundational models. (a) Data-effective learning aims to obtain a compact small dataset from a large-scale pre-training dataset, but the two datasets have similar effects on foundation model pre-training. (b) Demonstration of our comprehensive benchmark for DEL. The benchmark includes a dataset of millions of data samples from 31 medical centers (DataDEL), a baseline method for comparison (MedDEL), and a new evaluation metric (NormDEL).**

**Table 1: Overview of the proposed DataDEL in our benchmark. We integrate 31 medical centers, including data on the scale of millions, to build a high-quality, large-scale, multi-disease comprehensive dataset, aiming to provide researchers with a better data platform.**

| Division | Dataset | Task | Medical centers | Videos | Frames | Disease |
|---|---|---|---|---|---|---|
| Model Pre-training | Gastrovision [29] | Detection and classification | 2 | None | 8,000 | GI |
| | Hyper-Kvasir [8] | Classification | 1 | 374 | 110,079 | GI |
| | Kvasir-Capsule [52] | Classification | 1 | 117 | 47,238 | GI |
| Model downstream testing | CVC-12k (CVC-ClinicDB) [6] | Segmentation | 1 | None | 612 | polyp |
| | CVC-300 [55] | Segmentation | Not mentioned | None | 60 | polyp |
| | CVC-ColonDB [7] | Segmentation | Not mentioned | None | 380 | polyp |
| | EAD2019 [5] | Endoscopic artifacts | 6 | None | 2,991 | cancers |
| | EDD2020 [3] | Segmentation, detection, and localization | 4 | None | 380 | polyp |
| | ETIS [51] | Segmentation | Not mentioned | None | 196 | polyp |
| | ImageCLEFmed [27] | Polyp segmentation | 1 | None | 66,662 | polyp |
| | Kvasir-Instrument [28] | Segmentation, detection, and localization | 1 | None | 590 | polyp |
| | Kvasir-SEG [31] | Segmentation, detection, and localization | 4 | None | 1,000 | polyp |
| | Kvasir-Sessile [30] | Segmentation, detection, and localization | 4 | None | 196 | polyp |
| | PolypGen2021 [4] | Polyp segmentation | 6 | None | 3,762 | polyp |

## 1 Introduction

The effectiveness of foundation models depends on the abundance of pre-training data, a notion seemingly supported by consensus: more pre-training data leads to enhanced model performance. However, is this assumption truly accurate? To explore this issue, we introduce the concept of data-effective learning, which plays a significant role in the field of medical data, accelerating foundation model training and reducing storage burden in the era of big data (Figure 1(a)). The global endoscopy surgery market is undergoing significant development, with an estimated daily addition of 22,546,800,000 video frames [1]. However, a substantial presence of disruptive and invalid data [45] significantly hampers training efficiency and occupies a considerable amount of storage space [60]. Therefore, achieving data-effective in endoscopy datasets holds the following special advantages:

- **Storage Savings**: Assuming the use of traditional high-definition endoscopes (1080p) [26], the daily uncompressed endoscopy examination videos would require 12,756,493 TB (about 13.06 exabytes) of storage space. Condensing endoscopy datasets can result in substantial storage savings.
- **Enhanced Model Efficiency**: [8] released the largest digestive system image dataset (Hyper-Kvasir) to date and showed that upon analyzing data sources, over 90% of video frames consist of disruptive and invalid data. Core critical data comprises only 2% of the entire dataset [30, 31]. Efficient utilization of core critical data can significantly improve the efficiency of model training, in order to achieve rapid convergence.
- **Computational Resource Savings**: With the use of a single RTX 3090 graphics card and the VGG16 model [33] (approximately 138 million parameters) for 32-bit floating-point (FP32) calculations, processing 325.6 images per second is achievable [35]. In this configuration, training on the daily added video frames would require 19,200 hours. Utilizing only core critical data could save nearly 18,816 hours without compromising precision.

The issue of data-effective learning is not exclusive to endoscopy datasets; it is also evident in other types of medical datasets. With the rapid expansion of future medical data, efficiently handling medical datasets is the next crucial research problem in data-driven learning methods[47].

In recent years, there has been relevant research on data-effective learning in natural image datasets such as adversarial samples [20], and data biases [56], aiming to enhance the robustness and generalization capabilities of deep learning models. These works in natural datasets have shown preliminary effectiveness, indicating significant potential advantages such as data storage, computational resource savings, and efficient model training. However, despite the exponential growth of medical datasets, there is currently a lack of such exploration in the medical field [58]. This is attributed to the absence of relevant benchmarks, encompassing unified datasets, baseline methods, and comprehensive evaluation metrics, providing the academic community with a basis for developing and comparing advanced methods.

In addition to the absence of benchmark, achieving data-effective in the field of endoscopy remains a huge challenge [50] and is worth exploring in depth. Traditional data-effective techniques may struggle to work effectively with datasets that have high-dimensional features or sparse data. In endoscopy datasets, the challenge lies more in identifying subtle differences in images, which can be crucial for medical diagnosis. Therefore, accurately determining the similarity of images in endoscopy datasets is a top priority in research. Maintaining data quality and integrity is a key issue in data-effective learning [18], as it directly affects the usability and accuracy of the dataset.

Our research contributions can be summarized as follows:

- We introduce the concept of data-effective learning and provide a corresponding medical benchmark (Figure 1(b)) to guide data-effective algorithm research in the medical field. Furthermore, we integrate an open-source dataset called DataDEL, sourced from a million-level dataset spanning 31 medical centers.

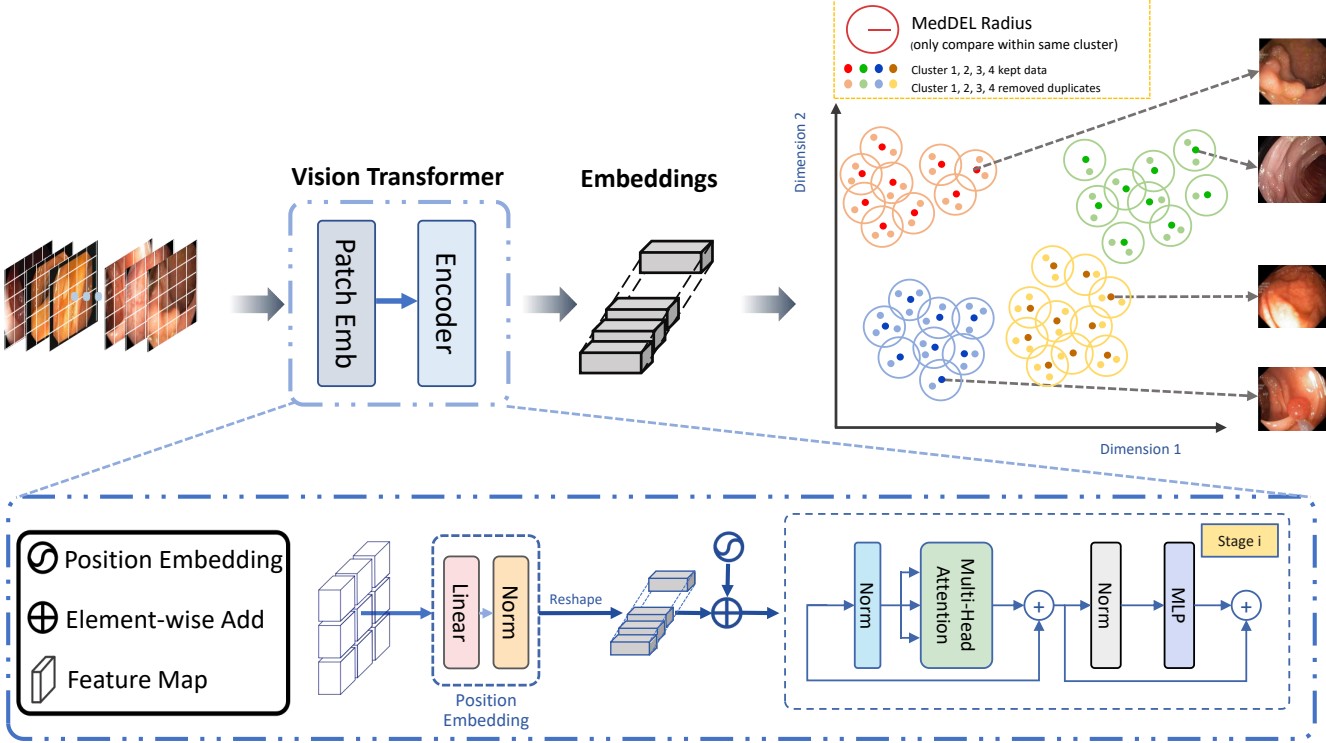

**Figure 2: Pipeline of the baseline method (MedDEL) for data-effective learning in our benchmark. It illustrates effective removal of disruptive and invalid data from the dataset, aiming to save storage space and computational resources while enhancing model efficiency.**

- In our benchmark, we introduce a baseline method called MedDEL for data-effective learning, which can outperform the use of 100% of the data in downstream tasks with 5% of the pretraining data in extreme cases.
- We develop a new metric called NormDEL, to assess the performance of data-effective in datasets, which considers the relationship between the proportion of the dataset retained and the performance of downstream tasks.

## 2  Related Work

In the natural image domain, benchmarks for assessing data-effective learning typically involve various techniques such as structural similarity index [17], convolutional neural networks [39], and local feature descriptors [36]. These methods [44, 57] are evaluated on widely used image datasets like ImageNet [15, 49] and CIFAR [34], or specialized datasets. PXDedup [59] explores the issue of data-effective learning in JPEG images, pointing out that traditional binary stream-based techniques do not work well for compressed JPEG images. [9] introduces a high-precision image data effective method that identifies and eliminates duplicate images through feature extraction and high-dimensional indexing [32]. [10] uses wavelet decomposition [43] to extract feature vectors from images and calculate the Manhattan distance to determine image similarity, thus achieving the detection and removal of duplicate images. However, when handling datasets with high-dimensional features [48] or sparse data [59], traditional data-effective techniques might not work effectively. MFDedup [61] surpasses existing technologies in improving data reduction rates and recovery throughput, while also reducing the cost of garbage collection. Storage and memory capacity [13] can also become limiting factors. [24] discusses how to accurately estimate the data reduction ratio achieved through data effective and compression techniques for specific datasets. Hash methods [46] indeed have a wide application in data effective processing. The CE-Dedup [38] framework combines hash-based image data-effective techniques with deep learning image classification tasks. By adjusting the deduplication threshold, effectively balances the trade-off between data reduction rate and model accuracy [21]. [40] proposes a deep supervised hash algorithm for image retrieval by providing the model with pairs of images labeled as similar/dissimilar. [14] proposed a Core-set selection method based on metric explanations (CSUME) for the classification of multi-class electrocardiograms. This work offers us an effective method for selecting representative datasets within an unsupervised learning framework, which holds significant importance in the fields of medical image computing and health informatics.

However, in the medical field, there currently exists no such benchmark for the research of data-effective learning algorithms.

## 3  Data-Effective Learning Medical Benchmark

In this section, we introduce the definition of benchmark tasks. Data-effective learning refers to the practice of using a limited amount of data for the pre-training phase [19]. Therefore, this benchmark

encourages researchers to develop advanced data-effective learning methods to generate a compact version of the originally collected large-scale dataset, thereby obtaining a compact small-scale new dataset for pre-training foundational models.

In the era of big data, large models often require massive pre-training data [11]. However, our perspective challenges this conventional thinking, and we have undertaken the following work to explore this issue in depth.

Our benchmark consists of three parts, a dataset with millions of data samples (DataDEL), a baseline method for comparison (Med-DEL) and a new evaluation metric (NormDEL).

### 3.1 The benchmark dataset: DataDEL

Given the urgent demand for a comprehensive benchmark in the medical field, we are facing the pressing task of integrating diverse large datasets. Currently, prevailing challenges within existing medical datasets encompass issues like the uniformity of data sources, the standardization of task execution, the limited scope of covered disease types, imbalances in dataset categories, and the uniformity of modalities [37].

These limitations hinder the comprehensive development of medical research. Therefore, there is an urgent need for us to construct a comprehensive medical dataset. Our efforts focus on integrating datasets from over 31 different centers and 23 different countries, spanning multiple modalities such as images and videos. Furthermore, our dataset surpasses the million-scale mark, becoming a large-scale collection that encompasses multiple tasks, modalities, sources, and diseases. This dataset can provide rich and diverse support for subsequent research, offering significant convenience for healthcare professionals and research institutions.

### 3.2 The benchmark baseline: MedDEL

We introduce a benchmark baseline method (MedDEL) in the endoscopic medical field, which is based on the principles of the SemDeDup method [2]. Endoscopic data holds significant importance in medical diagnosis and treatment. However, due to various factors such as organ morphological changes [53], lighting conditions [12], and noise interference [23], the analysis of this data becomes complex and challenging.

Specifically, the MedDEL method takes an image $I$ as an $i^{th}$ example, the encoder of Vision Transformer (ViT) [16] extracts deep features from the image, ultimately producing a 768-dimensional feature output. This output serves as our information embedding. The high-dimensional representation not only captures the semantic information of the image more effectively but also addresses semantic repetitiveness issues that are challenging to resolve in low-dimensional spaces. Considering the encoding process of image $I$ through the ViT model, we can represent it with the following formula:

$$F_x = ViT(I) \tag{1}$$

Where $x$ represents the final layer of the $ViT$ encoder.

After obtaining the feature embeddings for each image in the dataset, we can explore the similarity between images in a high-dimensional space. Let the embedding corresponding to the $i^{th}$ image be denoted as $F(x, i)$, representing all image features as

---

**Algorithm 1** Pseudo Code for MedDEL

**Input**: Image sequence $\{I_1, I_2, \ldots, I_n\}$
**Parameter**: Thresholds $\epsilon$ and $\eta$

1: Let $t = 0$
2: **while** $t < $ max_iterations **do**
3:     **for** $cluster_i \in clusters$ **do**
4:         **for** $j = 1$ to size($cluster_i$) **do**
5:             **if** $\text{dis}(F_{x,j}, centers_i) > \epsilon$ **then**
6:                 **Delete** $I_j$
7:             **else**
8:                 **for** $k = j$ to Size($cluster_i$) **do**
9:                     **if** $\cos(F_j, F_k) > \eta$ **then**
10:                         **if** $\text{dis}(F_j, C_i) > \text{dis}(F_k, C_i)$ **then**
11:                             **Delete** $I_k$
12:                       **else**
13:                             **Delete** $I_j$
14:                     **end if**
15:                 **end if**
16:             **end for**
17:         **end if**
18:         **end for**
19:     **end for**
20:     Increment iteration counter: $t \leftarrow t + 1$
21: **end while**

---

**Table 2: Experimental setting for the parameter of $\eta$ in Algorithm 1 that controls the data remaining ratio. We set several different ratios to fully assess the performance of MedDEL under various proportions of remaining data. Additionally, we also calculate the number of training epochs required for different ratios of data at the same computational power for fair comparison.**

| $\eta$ | Remaining data | ratio | Epochs |
|------|------|------|------|
| 1.0 | 88,282 | 100% | 200 |
| 0.9 | 43,484 | 50% | 400 |
| 0.85 | 28,352 | 33% | 600 |
| 0.8 | 16,789 | 20% | 1,000 |
| 0.75 | 9,055 | 10% | 2,000 |
| 0.7 | 4,461 | 5% | 4,000 |

$\{F(x, 1), F(x, 2), \ldots, F(x, n)\}$. For typical data-effective methods, we have to calculate the similarity between any two pairs of embeddings, resulting in an overall time complexity of $O(n^2)$. Taking our proposed upstream dataset as an example, which includes 88,282 images, the total number of calculations reaches 744 million.

However, by introducing the K-means algorithm [42], the overall computational complexity is reduced from $O(n^2)$ to $O(N^2/k)$, where $k$ is the number of clusters. After applying the K-means algorithm, we obtain $k$ clusters. Assuming that the feature $F(x, i)$ belongs to the $k$ cluster, we can derive the following $k$ sequences:

$$C_k = \{F(x, i) \mid i \in N^*, F(x, i) \in cluster_k\} \tag{2}$$

In each cluster, data points that are far from the cluster centroid are considered disruptive or invalid data. To filter out interference

and invalid data, we introduce a predefined threshold $\epsilon$. Specifically, in each cluster $C_j$ we define the distance from image $i$ to the cluster centroid as $d_{ij}$. By using the following inequality, we classify image $i$ as potentially affected by disruptive or invalid data.

Specifically, we introduce a similarity threshold $\eta$, which is used to define semantic duplicate information. In category $j$, we perform similarity calculations for each pair of embeddings using the following formula:

$$S(p, q) = cos(F(x, p), F(x, q)) \tag{3}$$

Through this similarity calculation, we retain a set of features that are closer to the cluster center, achieving the goal of filtering effective and core-important data. The pseudocode for a simplified algorithm regarding data effectiveness is presented in Algorithm 1.

### 3.3 The benchmark evaluation metric: NormDEL

Currently, there is no comprehensive objective metric for evaluating the performance of data-effective learning algorithms, where comprehensive refers to integrating both the test accuracy of downstream tasks and the compactness of pre-training data. Hence, We propose a new data-effective metric, the Normalized Data-Effective Learning Index (NormDEL). Our objective is that within the same task, if a model can achieve the same level of performance in downstream tasks using fewer pre-training data, its data-effective performance should be considered superior. Taking the commonly used segmentation task in endoscopy as an example, where the general metric for measuring performance is often mIoU (mean Intersection over Union), we can formulate the following equation:

$$DEL = mIoU \cdot e^{-\alpha \cdot R} \tag{4}$$

Where $\alpha$ serves as a positive weight parameter used to adjust the influence of mIoU and the retention ratio in DEL, and $R$ represents the proportion of the retained dataset.

Subsequently, we aim to normalize DEL to a range between 0 and 1 for a clearer definition of data-effective performance. Thus, we obtain the definition of NormDEL:

$$NormDEL = \frac{1}{1 + e^{-DEL}} \tag{5}$$

Through NormDEL, we can comprehensively evaluate the adaptability of the data-effective model with limited training data, emphasizing the critical factors of model data-effective in large-scale datasets.

## 4 Experiment

To assess the capability of MedDEL, we train the Foundation Model on three pre-training datasets as presented in Table 1. Subsequently, we conduct finetuning on downstream task datasets to evaluate performance.

### 4.1 Settings

To achieve the pretraining of the Foundation Model, enabling it to provide a better initial value for the majority of endoscopy tasks, we use the unsupervised Masked Auto Encoder (MAE-ViT-Base) [25].

During the training of all models, we selectively filter the pre-training dataset capacity based on different thresholds of $\epsilon$ and $\eta$. Specifically, we set $\epsilon$ to 0.9 and $\eta$ from 0.7 to 0.9 with a step size of 0.05. Consequently, we obtain the dataset quantities as presented in Table 2. The batch size is set to 16, and the training is conducted using AdamW [41] and a cosine learning rate [22]. The model maintains a peak learning rate of $1.5 \times 10^{-4}$ throughout the training process. In the validation of downstream tasks, we implement a simple Multi-Layer Perceptron (MLP) [54] to achieve segmentation heads.

### 4.2 Effective Data Utilization with MedDEL

Figure 3 demonstrates that, under equivalent computational resources, the MedDEL method which operates on a reduced dataset (utilizing only 5% of the data, indicated by the red curve), generally performs comparably to the full dataset (using 100% of the data, indicated by the black curve) across various downstream tasks. Moreover, in certain specific scenarios, due to its smaller dataset size and higher data quality, the MedDEL method exhibits superior performance when trained on just 5% of the data. This advantage is particularly pronounced, as seen in the case of CVC-300, highlighting its effectiveness in efficiently leveraging a small amount of high-quality data. This underscores the MedDEL method's characteristic focus and refinement during the learning process, enabling it to more effectively capture crucial task features, resulting in improved performance in low-data scenarios. This result emphasizes the significance of MedDEL method in enhancing data effectiveness, particularly in the era of big data when high-performance models are needed, showcasing its capability for effective data utilization.

### 4.3 Storage Saving Analysis

Table 3 presents the impact of different amounts of pre-training data on mIoU and NormDEL across 8 downstream datasets. It is worth noting, in the majority of tasks, the mIoU achieved with only 5% of the data is comparable to that achieved with 100% of the data. To clearly assess the influence of pre-training data set proportion on performance, we computed NormDEL for each proportion. The results indicate that in some tasks, although the mIoU attained with 5% of the data may not be the highest when considering dataset size comprehensively, its NormDEL reaches the highest.

Furthermore, To analyze and evaluate the accuracy and robustness of the baseline method under the same time performance, Figure 4 illustrates a box plot of different proportions of pre-trained data in various downstream datasets, based on the mIoU probabilities. Each box represents the predictive performance of the model during the several final epochs, employing the median, approximate quartiles, and the lowest and highest probabilities to intuitively display the level, spread, and symmetry of the mIoU distribution.

Figure 4 demonstrates a unique trend in the performance of models (measured by the median mIoU) across most downstream tasks, in relation to the increase in the volume of pre-training data. Initially, performance improves with an increase in data volume, reaching a peak, and then starts to decline as data volume continues to grow. Specifically, when a smaller volume of data is used (as in the case of the 5% data volume), despite the limited quantity, the average information content per image is higher, enabling the model

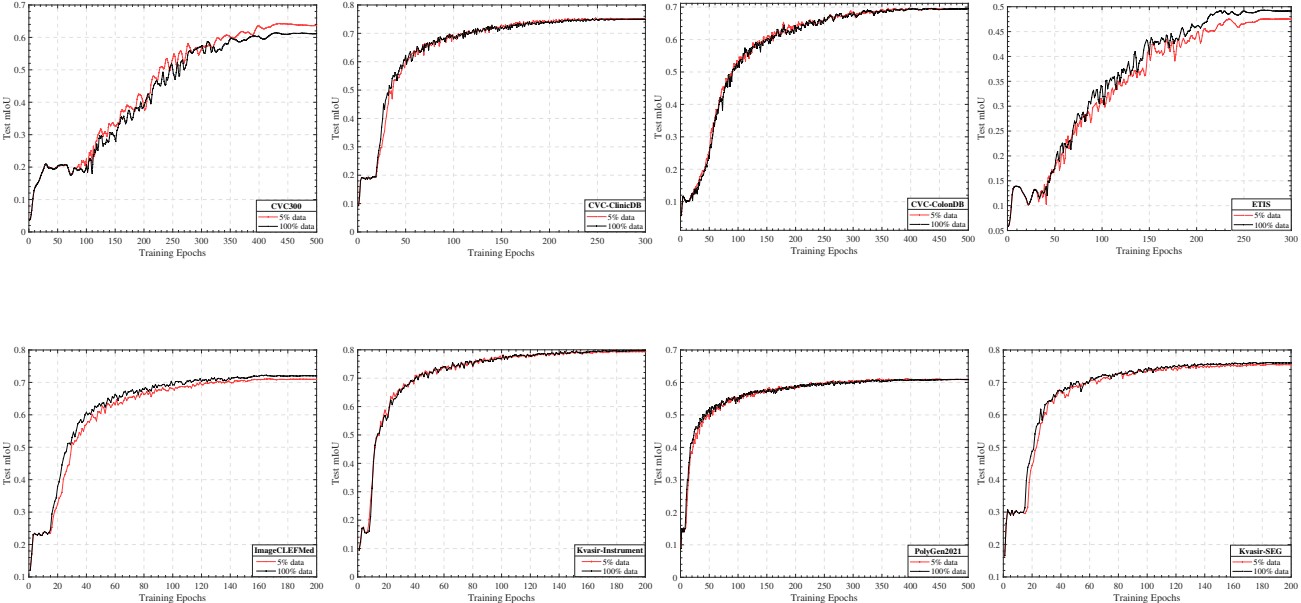

**Figure 3: Demonstration of the feasibility of data-effective learning. We compared the performance differences between using only 5% of the pre-training data (in red) and using 100% of the data (in black) in 8 datasets. The results indicate that using only 5% of the pre-training data can achieve results comparable to using 100% of the pre-training data, which fully demonstrates the validity of the MedDEL method.**

**Table 3: Demonstration of the rationality of NormDEL. The table shows the performance of different proportions of pre-trained data on mIoU (↑) as well as NormDEL (↑) in eight different datasets. The results indicate that the performance of mIoU is almost comparable in different scales, suggesting that it primarily measures performance without considering the scale of the pre-training data used. In contrast, NormDEL incorporates the factor of data scale, and can evaluate data compactness even with only 5% of the data used. This demonstrates the rationality of the NormDEL method, as it not only assesses performance but also effectively utilizes the data scale.**

| Dataset | 5% pretraining data | | 10% pretraining data | | 20% pretraining data | | 30% pretraining data | | 50% pretraining data | | 100% pretraining data | |
|---|---|---|---|---|---|---|---|---|---|---|---|---|
| | mIoU | NormDEL | mIoU | NormDEL | mIoU | NormDEL | mIoU | NormDEL | mIoU | NormDEL | mIoU | NormDEL |
| Kvasir-Instrument | 79.38 | **68.03** | 79.52 | 67.25 | 80.22 | 65.85 | 80.38 | 64.06 | **80.48** | 61.97 | 79.70 | 57.28 |
| Kvasir-SEG | 75.45 | **67.21** | 75.74 | 66.49 | 76.37 | 65.14 | 76.03 | 63.33 | **76.77** | 61.43 | 76.02 | 56.95 |
| ImageCLEFmed | 70.95 | **66.26** | 71.80 | 65.69 | 71.44 | 64.22 | **72.58** | 62.76 | 71.23 | 60.64 | 72.02 | 56.59 |
| ETIS | 47.50 | **61.11** | 49.44 | 61.00 | 50.20 | 60.13 | **50.28** | 58.94 | 49.62 | 57.47 | 49.13 | 54.51 |
| PolypGen2021 | 60.93 | **64.10** | 61.61 | 63.59 | 61.47 | 62.32 | **62.28** | 61.01 | 62.20 | 59.32 | 60.89 | 55.58 |
| CVC-300 | **63.67** | 64.69 | 56.69 | 62.55 | 61.40 | 62.31 | 62.85 | 61.11 | 61.97 | 59.29 | 61.16 | 55.60 |
| CVC-ClinicDB | 75.24 | **67.16** | 74.58 | 66.26 | 75.27 | 64.94 | **75.50** | 63.25 | 75.49 | 61.25 | 74.95 | 56.85 |
| CVC-ColonDB | 69.52 | **65.96** | 68.58 | 65.03 | 69.90 | 63.93 | **71.60** | 62.59 | 69.72 | 60.42 | 69.48 | 56.36 |

to achieve relatively decent results. However, as the data volume increases, redundancy in information also grows. At a certain point, a balance is reached between information content and redundancy, culminating in peak model performance. Beyond this point, further increases in data volume lead to a decline in performance. This trend is reflected in the length of the boxes in the plot, where the box length first increases with the data volume, reaches a peak, and then starts to shorten.

However, in the results from Figure 4, the trends are not as stable, and there is fluctuation observed in some of the downstream tasks.

This also reflects the limitations of the MedDEL method. These fluctuations may arise from differences in data characteristics, task complexities, or other factors. It is precisely this diversity that complicates the optimization of data effectiveness, requiring in-depth research and fine-tuning.

Nevertheless, these findings still underscore the effectiveness of MedDEL in reducing storage burden and achieving good results with smaller proportions of data. We encourage more researchers to engage with our study and propose more refined methods to make the most of limited data resources.

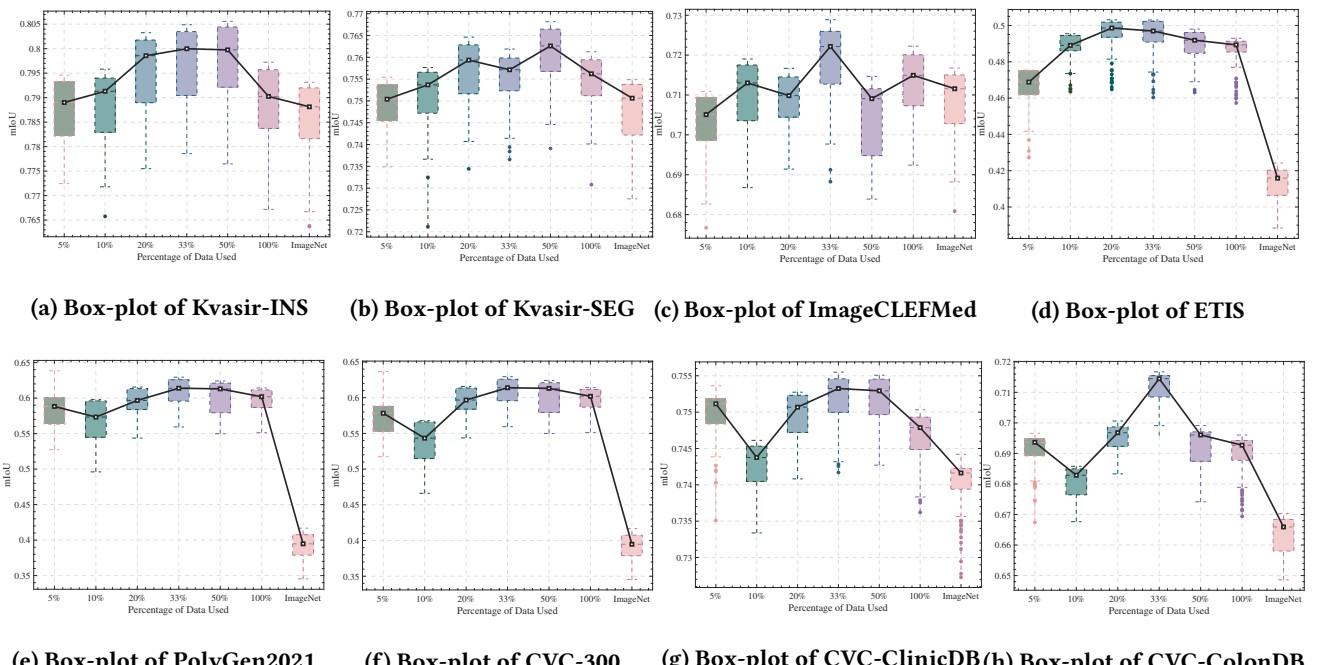

(a) Box-plot of Kvasir-INS        (b) Box-plot of Kvasir-SEG        (c) Box-plot of ImageCLEFMed        (d) Box-plot of ETIS

(e) Box-plot of PolyGen2021        (f) Box-plot of CVC-300        (g) Box-plot of CVC-ClinicDB        (h) Box-plot of CVC-ColonDB

Figure 4: Box plot of the baseline method (MedDEL) for testing mIoU under retaining different data ratios. Under the same computational resources, we plot the impact of varying proportions of pre-training data on downstream task performance, using the data proportions from Table 2 and models pre-trained on ImageNet for reference. The results demonstrate that the performance of downstream tasks initially improves with an increase in pre-training data but eventually decreases. This is because the initially added data is effective, however, as more data is introduced beyond a certain point, it includes redundancy or erroneous information, which adversely affects the model's performance.

Table 4: Illustrating the impact of different pre-training time (epochs) on performance metrics (mIoU) using 20% and 33% of pretraining data. The results indicate that the optimal performance is often not achieved with the standard pretraining duration (1,000 epochs for 20% and 600 epochs for 33%) but rather occurs with early pretraining time, demonstrating the effectiveness of MedDEL in reducing computational resource consumption.

| Pretraining Time / Dataset | 20% pretraining data | | | | | | | | 33% pretraining data | | | | | |
|---|---|---|---|---|---|---|---|---|---|---|---|---|---|---|
| | 200 | 300 | 400 | 500 | 700 | 800 | 900 | 1,000 | 240 | 360 | 420 | 480 | 540 | 600 |
| Kvasir-Instrument | 80.21 | 79.73 | 79.98 | 80.01 | 80.06 | 80.11 | **80.22** | 80.22 | 80.25 | **80.50** | 80.32 | 80.39 | 80.41 | 80.38 |
| Kvasir-SEG | 75.80 | 75.69 | 76.13 | 75.90 | 76.25 | 76.33 | **76.39** | 76.37 | 75.73 | 75.94 | 75.84 | **76.07** | 75.78 | 76.03 |
| ImageCLEFmed | 71.44 | 71.60 | 71.44 | **72.91** | 71.51 | 71.52 | 71.84 | 71.44 | **72.59** | 72.32 | 72.34 | 72.54 | 72.19 | 72.58 |
| ETIS | 49.07 | 49.70 | 50.88 | 51.11 | 50.96 | **51.13** | 51.01 | 50.20 | 49.54 | 51.23 | **51.30** | 50.28 | 50.79 | 50.20 |
| PolypGen2021 | **61.64** | 61.32 | 61.54 | 61.00 | 61.47 | 61.15 | 61.30 | 61.47 | 61.15 | 61.62 | 62.10 | 62.28 | **62.29** | 62.28 |
| CVC-300 | 58.66 | 57.52 | 56.81 | 60.59 | **61.51** | 60.78 | 60.00 | 61.40 | 62.09 | 62.34 | **64.40** | 61.40 | 63.34 | 62.85 |
| CVC-ClinicDB | 74.66 | **75.56** | 75.48 | 75.29 | 75.11 | 74.83 | 75.41 | 75.27 | 75.44 | 75.11 | 75.24 | 75.17 | 75.28 | **75.50** |
| CVC-ColonDB | 69.83 | 70.17 | **70.68** | 69.65 | 70.62 | 70.10 | 69.89 | 69.90 | **72.09** | 71.62 | 70.89 | 71.43 | 71.95 | 71.60 |

## 4.4 Computing Power Saving Analysis

Section 4.3 demonstrates that MedDEL effectively enhances storage performance and achieves results comparable to large-scale datasets by using fewer data. Building upon this, we further explore the optimization of MedDEL's time performance. Table 4 presents models trained with 20% and 33% of pretraining data, using the epoch numbers from Table 2 as a reference. We progressively decrease the pretraining time in 10% increments, aiming to explore MedDEL's

performance metrics (mIoU values) across various downstream datasets at different pretraining times (epochs).

Remarkably, we observe that optimal performance is not consistently achieved with models trained using 100% of the pretraining time across nearly all datasets. Instead, we note the presence of a complex uncertainty and fluctuation, where the optimal result may be distributed among different pretraining time points in diverse datasets and with varying proportions of pretraining data. This underscores the effectiveness of MedDEL in conserving computational

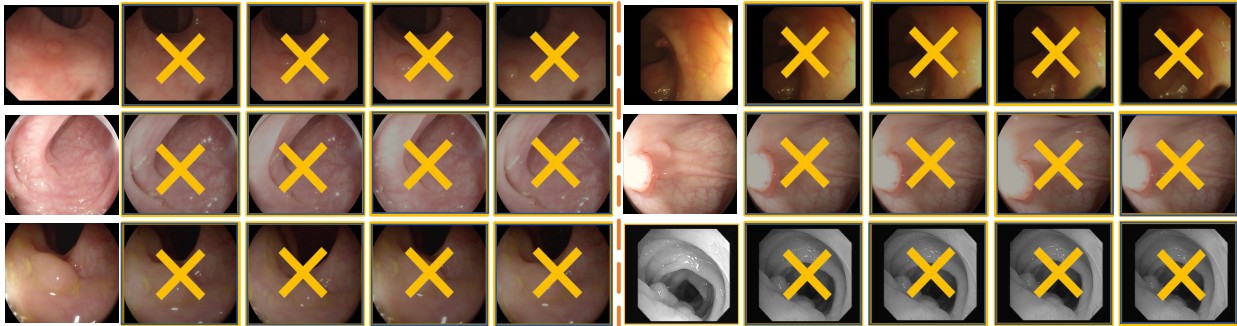

**Figure 5: Demonstration of images deleted by MedDEL. This figure shows MedDEL deleting semantically similar images, which appear to have no significant differences between them from a perceptual perspective,**

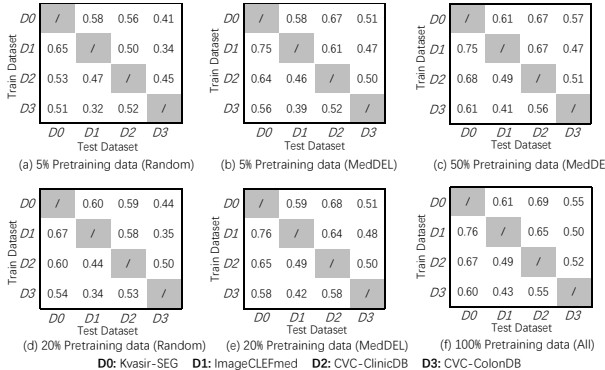

**Figure 6: Model generalization experiments across different datasets with different data volumes. The experiments included four distinct datasets: Kvasir-SEG, ImageCLEFmed, CVC-ClinicDB, and CVC-ColonDB. The results indicate the performance of the model at different volumes of pretraining data (5%, 20%, 50%, and 100%), and compared the outcomes between randomly selected data and data selected using the MedDEL method.**

resources. However, it also suggests that optimizing pretraining time may require different strategies for various tasks and data contexts. Our study has not yet uncovered the specific relationships between these strategies, indicating the need for more in-depth research in the future.

### 4.5 Visualization of MedDEL(DEL)

The MedDEL algorithm, as a benchmark method in the field of endoscopic, is based on the Vision Transformer (ViT) model [16] for extracting image features, generating a 768-dimensional feature output. By introducing K-means clustering [42], it successfully reduces computational complexity and further effectively addresses irrelevant data in the endoscopic dataset by setting similarity thresholds and filtering conditions.

Figure 5 visualizes the results of the MedDEL (DEL) algorithm, where upon observation, it is evident that MedDEL eliminates a

significant portion of redundant data with highly similar features from the dataset. Considering their similarity, we suppose removing such redundant data is a reasonable strategy to optimize the dataset structure, enhancing dataset quality. This optimization improves the model's understanding of endoscopic images, making it more targeted and interpretable, and providing robust support for medical image analysis [16].

### 4.6 Ablation Study

To validate the effectiveness of our model, we supplement the following generalization experiments in Fig 6. Specifically, we select 5%, 20%, and 50% of the data and compare the test performance of random selection and MedDEL. We find that whether using the random selection method or MedDEL, as the amount of data increases, the model's generalization ability improves. Furthermore, at the same data volume, MedDEL shows superior performance to the random method. Additionally, MedDEL shows comparable generalization performance when selecting 50% data compared to using full data.

### 5 Conclusion

In the era of foundation models, we are the first to propose whether larger pre-training data necessarily leads to improved model performance. In order to investigate this issue, we propose the first medical data-effective Benchmark, which involves a million-level dataset from 31 medical centers (DataDEL), a data-effective approach (MedDEL), and a comprehensive metric for evaluating pretraining data volume (NormDEL). The establishment of an open benchmark for data-effective learning is crucial for the medical artificial intelligence research community, which is an encouraging foundation for subsequent research endeavors.

### Acknowledgement

This work was supported in part by NSFC (No. U2001209, 62372117). The computations in this research were performed using the CFFF platform of Fudan University.

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
