# OpenReview forum: "A Medical Data-Effective Learning Benchmark for Highly Efficient Pre-training of Foundation Models"
_acmmm.org/ACMMM/2024/Conference — MM2024 Poster_

### Official Review · Reviewer_3jDs · 2024-05-22

**Rating:** 5
**Confidence:** 3

**Summary:**

The paper proposes a data-efficient learning benchmark for pre-training on medical data. The benchmark consists a dataset DataDEL, a baseline method for data-efficient learning (MedDEL) and a evaluation metric (NormDEL). Experiments and evaluations are performed to demonstrate the effectiveness of the benchmark.

**Strengths:**

1. The technical details are sound as demonstrated by Algorithm 1 and 4.1 Settings.
2. The experiments are performed on multiple datasets with reported results on different percentages of pre-training data.
3. Figure 3 shows the feasibility of data-efficient learning on multiple datasets with trends on training epochs.
4. In general, the paper provides many details on methodology and experiments with clarity and numerical results.

**Limitations:**

1. MedDEL is only compared with random selection in Table 5. Is there any other methods for data-efficient learning achieving the same purpose? If yes, how would them perform compared to MedDEL?

2. What is the purpose of DataDEL dataset? Is it a collection of existing datasets? It seems we can use proposed MedDEL and NormDEL to do the data-efficient learning on existing datasets, e.g. CVC-300, ETIS, as demonstrated in the experiments.

**Suitability:**

2

---

### Official Review · Reviewer_THsB · 2024-05-23

**Rating:** 4
**Confidence:** 4

**Summary:**

This paper proposes a benchmark for data-effective learning in the medical field, including a dataset with millions of data samples from 31 medical centers and a baseline model and A way to evaluate parameters and experimentally verify that the performance of the original data set can still be achieved using part of the data in the large data set.

**Strengths:**

1.Millions of data from 31 medical centers were collected to build a benchmark data set, covering a variety of diseases and tasks.
2.A simple but effective data-effective baseline learning method is proposed, and the K-means method is used to select the samples closest to the center point in different clusters.
3.Propose a new parameter for evaluating the relationship between the proportion of the data set used and the performance of downstream tasks

**Limitations:**

1.The baseline model only uses ViT as the encoder, which lacks diversity and does not take into account the adaptability of the pre-trained model itself to different amounts of data.
2.The downstream tasks only verify Segmentation and lack experimental verification for other tasks.

**Suitability:**

2

---

### Official Review · Reviewer_iZcF · 2024-05-26

**Rating:** 4
**Confidence:** 4

**Summary:**

The paper focuses on data efficient training of foundation models on healthcare. The large quantity of data requirement has proven to be a significant challenge in development of deep learning models in healthcare and thus the topic of research is highly relevant and of interest to broader community.  While the proposed approach is novel and the paper is well-written, certain limitations hinder the strength of the paper. The proposed dataset is extensive and would be a significant contribution to forward research, however this is not a dataset paper.  The author's claim "We introduce the concept of data-effective learning and provide a corresponding medical benchmark (Figure 1(b)) to guide data-effective algorithm research in the medical field" is not new as data efficient learning has been the focus on research for quite some time. The problem of core-set selection also translates to the same which has not been reviewed or discussed by the authors. The lack of comparison of the proposed approach with other techniques of data selection is missing with only random selection shown as comparison.

**Strengths:**

Comprehensive evaluation on multiple datasets.
Well-written
Novel Approach

**Limitations:**

While there may not be benchmarks on healthcare image datasets, there is a plethora of research on data efficient learning and core-set selection for healthcare data on various modalities of 2D images, 1D time-series signals and EHR tabular data. The lack of discussion of these works makes the related work not extensive enough.

Some Sample works:
1.Weicheng Kuo, Christian Häne, Esther Yuh, Pratik Mukherjee, and Jitendra Malik. 2018. Cost-sensitive active learning for intracranial hemorrhage detection. In the International Conference on Medical Image Computing and Computer-Assisted Intervention. Springer,715–723
2. S. Dakshit, B. M. Maweu, S. Dakshit and B. Prabhakaran, "Core-set Selection Using Metrics-based Explanations (CSUME) for multiclass ECG," 2022 IEEE 10th International Conference on Healthcare Informatics (ICHI), Rochester, MN, USA, 2022, pp. 217-225, doi: 10.1109/ICHI54592.2022.00041
3. Hossein Shokri Ghadikolaei, Hadi Ghauch, Carlo Fischione, and Mikael Skoglund. 2019. Learning and data selection in big datasets. In International Conference on Machine Learning. PMLR, 2191–2200

The proposed approach lacks comparison of other data efficiency approaches and core-set learning approaches which also have shown similar results. Without comparisons it is not possible to truly judge the efficiency or the benefit from the proposed approach.

**Suitability:**

3

---

### Official Review · Reviewer_vaWZ · 2024-05-28

**Rating:** 2
**Confidence:** 3

**Summary:**

The paper introduces a comprehensive benchmark, which includes a dataset (DataDEL) with millions of samples from 31 medical centers, a baseline comparison method (MedDEL), and a new evaluation metric (NormDEL). Experimental results demonstrate that the baseline method MedDEL can achieve performance comparable to that of the full dataset using only 5% of the original data.

**Strengths:**

1.	The paper integrate an open-source dataset, sourced from a million-level dataset spanning 31 medical centers.
2.	The paper conducted a large number of experimental verifications.

**Limitations:**

1.	In your MedDEL method, what is the innovation of the clustering method?
2.	Although methods based on feature extraction and high-dimensional indexing have been proposed for data deduplication, the ability to handle small differences in images still needs to be verified, especially in medical diagnosis, where these differences may be of important significance.
3.	In the medical field, model accuracy is crucial, so is it reasonable to reduce the amount of data at the expense of model accuracy? Specifically, in Table 3, if the NormDEL method is used, the best data set should be 5% pretraining data, but the highest mIoU is generally 30% pretraining data. There is a certain gap between the two mIoU. Whether your NormDEL method is too high depends on the proportion of data volume and ignores the accuracy of the model.
4.	If 5%, 20% and so on of the pretraining data are randomly selected for training, and then the NormDEL measurement standard is used to calculate the quality of the randomly sampled data set, will it still be concluded that the quality of the randomly sampled data is better than the full amount of data? Given this situation, how to explain the rationality of NormDEL and MedDEL?

**Suitability:**

3

---

### Meta-Review · Area_Chair_C48q · 2024-07-06

**Recommendation:** Accept (Poster)
**Confidence:** 4

**Metareview:**

This paper introduces a comprehensive benchmark for data-efficient learning in the medical field, including a large dataset, a baseline method, and a new evaluation metric. Reviewers acknowledge the paper's contributions, novelty, and clarity but raise concerns about the lack of comparison with other data efficiency approaches, limited discussion of related works, and the need for more extensive experiments. The paper's significance and impact on the multimedia/multimodal processing community are evident, but improvements are necessary to address the reviewers' concerns.

The paper's strengths outweigh its limitations. The authors have made significant contributions to the field, and the paper has the potential to inspire future research.